# Preparation and Characterization of Alginate Hydrogels with High Water-Retaining Capacity

**DOI:** 10.3390/polym15122592

**Published:** 2023-06-07

**Authors:** Ivana M. Savić Gajić, Ivan M. Savić, Zorica Svirčev

**Affiliations:** 1Faculty of Technology in Leskovac, University of Nis, Bulevar Oslobodjenja 124, 16000 Leskovac, Serbia; savicivan@tf.ni.ac.rs; 2Department of Biology and Ecology, Faculty of Sciences, University of Novi Sad, Trg Dositeja Obradovića 2, 21000 Novi Sad, Serbia; 3Faculty of Science and Engineering, Biochemistry, Åbo Akademi University, Tykistökatu 6A, 20520 Turku, Finland

**Keywords:** alginate, hydrogels, cross-linking, swelling, gelation time, water content, central composite design

## Abstract

Hydrogels are very attractive materials due to their multifunctional properties. Many natural polymers, such as polysaccharides, are used for the preparation of hydrogels. The most important and commonly used polysaccharide is alginate because of its biodegradability, biocompatibility, and non-toxicity. Since the properties of alginate hydrogel and its application depend on numerous factors, this study aimed to optimize the gel composition to enable the growth of inoculated cyanobacterial crusts for suppressing the desertification process. The influence of alginate concentration (0.1–2.9%, m/v) and CaCl_2_ concentration (0.4–4.6%, m/v) on the water-retaining capacity was analyzed using the response surface methodology. According to the design matrix, 13 formulations of different compositions were prepared. The water-retaining capacity was defined as the system response maximized in optimization studies. The optimal composition of hydrogel with a water-retaining capacity of about 76% was obtained using 2.7% (m/v) alginate solution and 0.9% (m/v) CaCl_2_ solution. Fourier transform infrared spectroscopy was used for the structural characterization of the prepared hydrogels, while the water content and swelling ratio of hydrogels were determined using gravimetric methods. It was concluded that alginate and CaCl_2_ concentrations play the most important role regarding the gelation time, homogeneity, water content, and swelling ratio of the hydrogel.

## 1. Introduction

Hydrogels are cross-linked hydrophilic polymers that swell in water and organic solvents but do not dissolve in them [1]. They can absorb a large amount of water, as well as substances that are soluble in them. Due to their multifunctional properties (high water content, soft and elastic consistency, etc.), hydrogels have been applied in agriculture, pharmacy, food, and cosmetic industries, water purification, contact lenses, sensors, etc. For the preparation of hydrogels, natural or synthetic polymers with hydrophilic groups are commonly used. According to the concept of green chemistry and environment protection, biopolymers are increasingly used to produce hydrogels, especially those directly extracted from natural sources, such as proteins and polysaccharides [2]. Unlike conventional polymers, natural polymers are biodegradable, biocompatible, and hydrophilic substances that require the use of eco-friendly solvents during gel preparation. In the focus of the scientific public, alginates still occupy an important place among polysaccharides.

Alginate is a water-soluble and linear anionic polysaccharide consisting of mannuronic acid (M-block) and glucuronic acid (G-block) [3]. It is mainly obtained by extraction from marine brown algae [4]. The most important property of alginate is the ability to form a gel in the presence of cross-linking agents, i.e., divalent cations, most often Ca^2+^ ions [5]. G-blocks are mainly responsible for binding divalent cations through the „egg-box“ model. The bonds of insufficient strength are formed between M-blocks and divalent cations.

Alginate hydrogels can be prepared using ionic cross-linking, covalent cross-linking, phase transition (thermal gelation), cell cross-linking, and free radical polymerization [6]. Among them, ionic cross-linking is the most used procedure for obtaining alginate hydrogels. According to this procedure, the hydrogel can be formed under very mild conditions and in the absence of toxic solvents. The properties of alginate hydrogel (porosity, swelling behavior, strength, stability, biodegradability, etc.) can depend on numerous factors. The structure and molecular mass of alginate, depending on the source and isolation procedure, have a significant impact on the hydrogel properties [7]. The gelation velocity, which depends on Ca^2+^ ions, has an impact on the uniformity and strength of the hydrogel [5]. At lower temperatures, the reactivity of Ca^2+^ ions decreases, which leads to a decrease in the gelation velocity. The mechanical properties of the hydrogel are improved by reducing the gelation velocity. The ionic cross-linking procedure at room temperature is recommended for hydrogel preparation because of lower energy consumption. Ramdhan et al. [5] studied the effects of the pH value of alginate and CaCl_2_ solutions on gel strength and the syneresis phenomenon. They reported that the increase in pH value of the alginate and CaCl_2_ solutions resulted in a decrease in the gel strength and syneresis degree. Alginate and Ca^2+^ ion concentrations also affect the hydrogel properties. The cross-linking interactions between the polymer chains and Ca^2+^ ions, as well as the strength of the gel structure, increase with increasing alginate concentration. The excessive concentration of alginate can reduce the diffusion of Ca^2+^ ions through the matrix, affecting the reduction of the possibility of cross-linking polymer chains. However, no significant variations in the gel structure were observed for the alginate concentration range of 1.3–1.7% [8]. The separation of alginate chains has occurred at high concentrations of Ca^2+^ ions, leading to gel disintegration. The effect of alginate and Ca^2+^ ion concentrations on the properties of hydrogels is important to study because of their further application. High water content, soft consistency, non-toxicity, low cost, and simple preparation enable a broad application of alginate hydrogels in the food, pharmaceutical, and cosmetic industries, as well as in agriculture, medicine, water purification systems, etc. [9]. The use of alginate hydrogels as a nutrient medium for bacteria growth is also significant [10].

The definition of optimal content of hydrogel with adequate properties represents hard labor work because it involves a lot of trial and error to achieve the final aim [11]. The one-factor-at-a-time (OFAT) method requires a higher utilization of the initial materials and consumed time compared to the advanced mathematical tool called experimental design. The experimental design is an experimental plan necessary for content optimization and evaluating the factors’ effect on the system response [12]. Unlike the OFAT, the experimental design approach can consider the effects of factors simultaneously on the defined system response [13]. Based on this analysis, the interactions between the analyzed factors can also be determined. The experimental design is related to response surface methodology (RSM), which is responsible for generating mathematical models to predict the defined response with high accuracy. The global optimal conditions can be reached according to the RSM, while the OFAT method gives the local optimal conditions. The second-order polynomial equation is commonly utilized for process optimization [14]. For that purpose, the central composite design (CCD) with this polynomial equation is an adequate experimental design that can be used. This design consists of cube, axial, and center points. The cube points represent the combination of −1 and +1 levels. The axial plots need to predict the behavior of the proposed model in the extreme conditions/values of all analyzed factors (out of design space for all factors) so that these experiments are the combination of ±α and 0 (medium) levels. The center point is repeated to obtain the statistics of fitting data. This design point represents the combination of the mean values of the factor levels. Yang et al. [15] developed and optimized the composition of a Poloxamer 407-based floating hydrogel using the CCD. The concentrations of Poloxamer 407 and sodium bicarbonate were analyzed as the factors. The optimal composition of the gelatin-*κ*-carrageenan polyelectrolyte complex for the development of extended-release dosage forms was obtained using rotatable CCD after varying the content of gelatin and *κ*-carrageenan [16]. Mishra et al. [17] used the CCD with two factors and three levels to optimize, formulate, and characterize the sulfasalazine-loaded lipoidal nanoparticle-based hydrogel. In that case, the amount of stearic acid and Tween 80 was varied to obtain the optimal content of the formulation.

Our research team aims to find an innovative, integrated, and sustainable solution that relies on problems related to damaged land surfaces, desertification, and air/water pollution. It has been suggested that rehabilitation by cyanobacterial inoculation could reverse the damage to land surfaces, desertification processes, and associated problems. One of the main problems is the lack of moisture in the initial growth phase of inoculated cyanobacteria. Considering these facts, the plan was to use alginate hydrogel with a high water-retaining capacity (WRC) for cyanobacterial growth. This study aimed to optimize the procedure of alginate hydrogel preparation with adequate mechanical properties. The alginate concentration was analyzed in the range of 0.1–2.9% (m/v), while the Ca^2+^ ion concentration was analyzed in the range of 0.4–4.6% (m/v). Fourier transform infrared spectroscopy (FTIR) was used for the structural characterization of the prepared hydrogels. In order to ensure the use of prepared hydrogels as the carriers for cyanobacteria, the effects of alginate and Ca^2+^ ion concentrations were analyzed on their gelation time, homogeneity, water content, and swelling ratio (SR).

## 2. Materials and Methods

### 2.1. Chemicals and Reagents

In this study, calcium chloride dihydrate and alginic acid sodium salt—very low viscosity (pH 5.68 and dynamic viscosity of 37.8 mPa·s) were purchased by Thermo ScientificTM (Waltham, MA, USA). Other used chemicals were *p.a.* quality.

### 2.2. Preparation of Alginate Hydrogel by Ionic Gelation

The hydrogel was prepared by an ionic gelation process based on the interaction between alginate and divalent ions (calcium ions) from the CaCl_2_ solution. Before crosslinking with divalent ions, alginate solutions were stirred in water at room temperature for 24 h. The alginate solution was placed into a 100 mL plastic syringe and then dropwise at a constant flow through a 26 G (0.45 × 12 mm) blunt-tipped metal needle into the gelling solution (CaCl_2_ solution). The CaCl_2_ solution was homogenized using the magnetic stirrer. In this way, the spherical particles can be prepared. The CCD with two factors was used to investigate the effect of alginate and CaCl_2_ concentrations on the WRC of alginate hydrogels. These factors varied in the design space, so the thirteen formulations were necessary to prepare according to the design matrix. The coded and actual values of the five-factor levels are depicted in Table 1.

The obtained WRC of alginate hydrogels was fitted using different polynomial equations. The mathematical models were generated and statistically processed in the software package Design-Expert 13 (Stat-Ease, Minneapolis, MN, USA). Analysis of variance (ANOVA) was used to assess the statistical significance of the terms in the polynomial equation. Furthermore, the polynomial models were compared to each other based on the coefficient of determination.

### 2.3. Determination of Water-Retaining Capacity

The WRC of alginate hydrogel represents the ability of gel structure to retain water. It was determined based on the following procedure: gel was cut in half and centrifuged at 6000 rpm for 10 min. After centrifugation, the water was decanted from the hydrogel sample. The hydrogel masses before and after centrifugation were measured and then used to calculate the WRC of alginate hydrogel according to Equation (1) [18]:(1)$$WRC\left(\%\right)=\frac{W_{a}}{W_{b}}\times 100$$
where *W_a_* and *W_b_* are the hydrogel masses before and after centrifugation, respectively.

### 2.4. FTIR Analysis

The FTIR spectra of the dried formulations were recorded after the preparation of KBr pellets. The samples were recorded in the wavenumber range of 4000–400 cm^−1^ with a resolution of 2 cm^−1^ on a FT-IR spectrophotometer (Bomem MB Serie Hartmann & Braun-Michelson, MB series, Quebec, Quebec, Canada). After recording, the spectra were processed using Win-Bomem Easy software (Bomem GRAMS/32, Galactic Industries, Salem, NH, USA).

### 2.5. Determination of Gelation Time

The gelation time, which represents the time between the alginate addition and the hydrogel formation, was determined as follows: the container in which the hydrogel was formed was periodically tilted during the gelation process. The hydrogel was formed when it no longer flowed in the case when the container was tilted at an angle of 45° for more than 30 s.

### 2.6. Determination of Hydrogel Homogeneity

The homogeneity of the alginate hydrogel was evaluated based on the following way: the hydrogels were cut into four parts of equal dimensions (a–d) [19], which were measured before and after drying. The mass ratio of the samples after and before drying (m_d_/m_w_) of the hydrogel parts represents the homogeneity. A homogeneous gel had approximately the same mass ratio.

### 2.7. Determination of Water Content in Alginate Hydrogel

The moisture content of hydrogel was determined gravimetrically. The prepared hydrogel (1 g) was measured and dried at 40 °C in a laboratory oven for 3 h. After drying and cooling in a desiccator, the sample mass was measured. The procedure was repeated three times with a drying time of 30 min. The water content was calculated according to Equation (2):(2)$$Water\;content\left(\%\right)=\frac{m_{1}-m_{2}}{m_{1}}\times 100$$
where *m*_1_ is the mass of the sample before the drying process (g), and *m*_2_ is the mass of the sample after the drying process (g).

### 2.8. Determination of the Swelling Ratio of Alginate Hydrogel

The SR of alginate hydrogel formulations (FH1-FH13 and optimal formulations) was monitored gravimetrically. The dried alginate hydrogels (0.05 g) were poured with 20 mL of distilled water and mixed on a magnetic stirrer at room temperature. In certain time intervals (0.5, 1, 2, 4, 6, 8, and 24 h), the sample mass was measured after decanting the liquid medium and drying it with filter paper.

The SR (*α*) of alginate hydrogel was calculated according to Equation (3):(3)$$\alpha \left(\frac{g}{g}\right)=\frac{m_{t}-m_{0}}{m_{0}}$$
where *m_t_* is the mass of swollen particles at time *t*, and *m*_0_—is the initial mass of dry particles.

### 2.9. Statistical Analysis

All experiments were conducted in triplicate. The results were expressed as the average value with the standard deviation. One-way ANOVA with Tukey’s HSD post hoc test generated in IBM SPSS Statistics software (version 25, Chicago, IL, USA) was used to estimate the significant differences among samples. The samples with a *p* < 0.05 were considered significantly different. 

## 3. Results and Discussion

### 3.1. Modeling of Alginate Hydrogel Preparation Procedure

For the preparation of hydrogels, an aqueous solution of alginate and an ionic cross-linking procedure were used. This procedure is convenient because the hydrogel can be formed at room temperature in the absence of toxic solvents. These conditions are important due to the minimized cell death and maintaining the overall cellular function of cyanobacteria. The effects of alginate (0.1–2.9%, m/v) and CaCl_2_ (0.4–4.6%, m/v) concentrations on the WRC of alginate hydrogels were analyzed based on the CCD matrix (Table 2). The stoichiometric ratio of 1:2 between alginate and Ca^2+^ ions was constant in all experiments. The central point of the design was repeated five times to obtain the statistical parameters of the regression model. The WRC of alginate hydrogel was in the range of 18.7–73.2%. The lowest value was obtained for the hydrogel formulation prepared using 0.1% (m/v) alginate solution and 2.5% (m/v) CaCl_2_ solution. The highest WRC of alginate hydrogel was noticed for the formulation obtained from 0.5% (m/v) alginate solution and 1% (m/v) CaCl_2_ solution.

The obtained data were fitted using polynomial equations of different orders (linear, 2FI, quadratic, and cubic equation), which statistical data are given in Table 3. The selection of a suitable model was made based on the maximum value of the adjusted and predicted coefficient of determination. The quadratic model was taken into consideration for further analysis since the coefficient of determination had the highest value.

The other statistical data for the quadratic model are depicted in Table 4. The coefficient of determination of 0.965 indicates that the suggested regression model can explain 96.5% of the variation in the WRC of alginate hydrogel. An adequate precision of 20.99 indicated a good signal-to-noise ratio since it should be greater than 4 [20]. The coefficient of variation was 7.5%. A small value indicated that the developed model could predict the system response quite accurately [21]. Moreover, the difference between the adjusted and predicted coefficient of determination was acceptable, i.e., lower than 0.2.

The polynomial equation in terms of codded factors which is useful for identifying the effect of factors by comparing their regression coefficients is presented in Equation (4). Among the linear terms, the alginate and CaCl_2_ concentrations had a positive effect on the WRC of alginate hydrogel. The effect of alginate concentration on the system response (*Y*) was the highest compared to the other factors.
(4)$$Y=49.4+22.4A+3.68B-16.9AB-8.03A^{2}+8.6B^{2}$$

ANOVA was used to assess the statistical significance of the terms in the polynomial equation at a 95% confidence level (Table 5). 

The sum of squares (SS), degrees of freedom (df), mean square (MS), *F,* and *p*-value were calculated. Of all linear and quadratic terms, only the linear term of CaCl_2_ concentration was not statistically significant. The model was reduced by excluding the statistically insignificant terms to improve its prediction ability [20]. The *F*-value of the model (38.87) was statistically significant, while the *F*-value for lack-of-fit was statistically insignificant relative to pure error [22].

In Figure 1a, the dependency between predicted and actual values of the WRC of alginate hydrogel is depicted. Since the data follow a straight line, it can be concluded that there is a good agreement between the experimental and predicted WRC of alginate hydrogel. The resulting S-shaped normal probability plot (Figure 1b) indicated less variance than expected [23]. Cook’s distances were less than the threshold value (0.983), so this model did not have outliers in the design space.

The perturbation plot is suitable for analyzing the effect of individual factors on the defined system response. The effect of alginate concentration was statistically significant compared to CaCl_2_ concentration (Figure 2a) which was according to the results of ANOVA and regression coefficients. A strong interaction was noticed between the alginate and CaCl_2_ concentrations, which was derived based on the intersection of the lines of the effects (Figure 2b).

The three-dimensional plots were software generated to more easily consider the factor effects on the system response (Figure 3). A statistically significant increase in the WRC of alginate hydrogel occurred with increasing alginate concentration, but only at the lower CaCl_2_ concentrations. The WRC of alginate hydrogel increased at lower alginate concentrations, while its value decreased with increasing CaCl_2_ concentrations.

### 3.2. Optimization of the Alginate Hydrogel Composition

The desired alginate hydrogel composition was determined using a numerical optimization method. During the optimization, the WRC of alginate hydrogel was maximized in the concentration range of 1.5–2.5% (m/v) for alginate solution and 1–4% (m/v) for CaCl_2_ solution. The optimal composition of alginate hydrogel was achieved using 2.7% (m/v) alginate solution and 0.9% (m/v) CaCl_2_ solution. Under the proposed optimal composition, the regression model predicted 75.8% of WRC for alginate hydrogel. In order to check the validity of the proposed regression model, the hydrogel was prepared under the given composition. The WRC of thus prepared alginate hydrogel was 76.0%, which was close to the predicted value. This behavior of the regression model indicated its adequate prediction ability of WRC for alginate hydrogel. The optimal composition of alginate hydrogel with the desirability function is depicted in Figure 4. Since the prepared formulation could retain water, it could be used for growth support of cyanobacteria [24]. Since formulations 1, 2, 7, 11, and 12 represent the repeated experimental runs of the design center point to obtain the statistical data of the proposed models, one of these formulations was chosen and subjected to further characterization. Due to that, the F1 formulation was selected as suitable for FTIR analysis, determination of gelation time, homogeneity, water content, and SR.

### 3.3. FTIR Analysis for Structural Characterization of Alginate Hydrogel

FTIR spectra of sodium alginate before and after cross-linking with Ca^2+^ ions and formed hydrogels are presented in Figure 5. In the spectrum of sodium alginate (Figure 5a), a broad, intense band at 3452 cm^−1^ was the result of valence vibrations of the OH bonds. The valence vibrations of the aliphatic C–H bonds were detected at 2974 and 2928 cm^−1^. The bands at 1608 and 1439 cm^−1^ can be attributed to asymmetric and symmetric valence vibrations of the carboxylic anion (COO-). The valence vibrations of the –C–O–C– bonds for ether and alcohol groups had the bands at 1093 cm^−1^ and 1040 cm^−1^, respectively.

After cross-linking of alginate with Ca^2+^ ions (Figure 5b–k), substantial changes in the wavelength number at 1608 and 1439 cm^−1^ were noticed for asymmetric and symmetric valence vibrations of the carboxyl anion, respectively. These bands were shifted to higher wavelengths (from 1608 to 1651 cm^−1^ and from 1439 to 1458 cm^−1^) as a result of the strong electrostatic interaction between the Ca^2+^ ions and the carboxyl group of alginate. After alginate cross-linking, the changes in the band intensity (mainly a decrease) originating from the stretching vibrations of the –C–O–C– bond of the ether and alcohol groups were also observed. These characteristic bands of sodium alginate and calcium alginate particles are also described in the available literature [25].

### 3.4. Gelation Time, Homogeneity, and Water Content of Alginate Hydrogel

Gelation time is an important parameter for the estimation of the hydrogel’s mechanical properties. It mainly depends on the concentration of released Ca^2+^ ions from the used salt particles. For example, the gradual release of Ca^2+^ ions from CaCO_3_ particles impacts the extension of gelation time. Because of that, the formed hydrogel is soft and has a homogeneous structure [26]. In Table 6, the gelation time of FH1–FH13 and optimal formulations are ranged between 12 and 22 min. The shorter gelation time was observed for the formulations prepared at higher concentrations of alginate and Ca^2+^ ions (FH3, FH5, and FH6 formulations). The longest gelation time was noticed for the FH4 and FH9 formulations, which were obtained at lower concentrations of alginate and Ca^2+^ ions. It is well known that a shorter time is undesirable since it may result in more difficult shaping and decreased gel uniformity. In the literature, the gelation time of alginate hydrogel can be from a few minutes (about five minutes) [19] and even up to several hours (about two hours) [27]. Such a large deviation in the gelation time can be attributed to the difference in the hydrogel composition. By analyzing and comparing the gelation times obtained in this study with those available in the literature, it can be concluded that the optimal composition formulation had an adequate gelation time of 19 minutes. This hydrogel also had an adequate consistency, which could be used as a support for the growth of cyanobacteria.

The mass ratio of the dry and wet parts (a–d) of the prepared formulations is a measure of their homogeneity (Table 6). Since the mass ratios for all four parts of the analyzed samples were almost the same, the prepared hydrogels can be considered homogenous. A decrease in homogeneity was observed for formulations with a higher Ca^2+^ ion content (FH3, FH5, and FH13 formulations). With increasing the alginate content in the hydrogel formulation, there was an improvement in homogeneity (FH4, FH6, and FH10 formulations), which was also noted in the optimal formulation. This fact is consistent with the findings of other authors who also analyzed the homogeneity of alginate hydrogel [18].

The water content in the alginate hydrogel was in the range of 92.69–99.77% (Table 6). The FH3 formulation had the lowest water content, while the FH5 formulation had the highest water content. The difference between them was statistically significant at the 95% confidence level. Based on the obtained water content, it can be said that the concentrations of alginate and Ca^2+^ ions have a significant impact. The water content increased at higher concentrations of Ca^2+^ ions and lower concentrations of alginate. In the optimal hydrogel formulation, the water content was 94.26%, which is considered favorable for cyanobacteria growth. The water content of alginate hydrogel formulations is consistent with that described in the literature (>75%) [28].

### 3.5. Swelling Study

One of a hydrogel’s key characteristics is its swelling capacity, which reveals information about the hydrophilicity of the polymer network and the relative density of the cross-linking. Tighter-networked hydrogels typically exhibit less SR. Swelling properties may be used to identify batch-to-batch changes and consistency in the production quality of hydrogels, as well as to determine whether their mechanics are changing over time. In addition to the optimal formulation, the SR of the other prepared formulations was determined. The swelling of the hydrogels was monitored over time by the gravimetric method. The swelling kinetics curves of the prepared hydrogel formulations are depicted in Figure 6.

The SR of the prepared formulations was increased rapidly during the first two hours (phase of rapid water binding), followed by a short plateau and a second phase in which SR increases sharply. With all formulations, it was observed that the equilibrium phase was reached after eight hours. Generally, swelling is related to the cross-linking degree, i.e., its value is significantly influenced by the concentration of Ca^2+^ ions. The FH4 and FH10 formulations had a higher SR compared to the other observed formulations since the concentration of Ca^2+^ ions was low (0.4% for the FH4 formulation and 1% for the FH10 formulation). Therefore, the density of cross-linking was low for the optimal formulation (the maximum SR of 9.40 g/g). In contrast, the higher concentration of Ca^2+^ ions in FH3 and FH13 formulations leads to a greater cross-linking of alginate chains and a decrease in the WRC of a hydrogel. For all hydrogel formulations, no significant mass loss was observed after 24 h of incubation (data not shown), which indicated that the stability of alginate chains was improved by cross-linking with Ca^2+^ ions. Li et al. [25] also confirmed that the SR depends on the concentration of Ca^2+^ ions. The swelling study indicated that the alginate hydrogel mechanics are dictated by the Ca^2+^ ions concentration. This is coupled with changes in the network density, which limits independent control of these features. The high SR of optimal formulation indicated a larger mesh size or molecular porosity. Such hydrogel structure should ensure a better flow of nutrients through the matrix and therefore ensure the survival of cyanobacteria in the initial growth phase.

## 4. Conclusions

In this study, alginate hydrogel was prepared in the absence of toxic solvents at room temperature by ionic gelation. Ca^2+^ ions were used for cross-linking alginate chains. The formulation composition was optimized to obtain the maximum WRC of alginate hydrogel. The CCD was applied to analyze the effect of alginate (0.1–2.9%, m/v) and CaCl_2_ (0.4–4.6%, m/v) concentrations on the WRC of alginate hydrogel. The best fitting of data was achieved using the reduced polynomial equation with a coefficient of determination of 0.994. The ANOVA analysis showed that only the linear term of the CaCl_2_ concentration in the polynomial equation was not statistically significant. The hydrogel formulation with the maximum WRC of alginate hydrogel was achieved with the following composition: 2.7% (m/v) of alginate solution and 0.9% (m/v) of CaCl_2_ solution. Based on the good agreement between the experimental and predicted WRC of alginate hydrogel, the proposed optimal conditions can be considered valid. The prepared hydrogel formulations were subjected to the analysis of their physical and chemical properties. The optimal formulation, like other hydrogel formulations, retains good homogeneity and high-water content. The SR was significantly influenced by the concentration of Ca^2+^ ions. The highest SR was determined for the formulation with the lowest concentration of Ca^2+^ ions. The optimal formulation had an SR of 940.0%. In summary, the optimal composition of alginate hydrogel was developed for potential application as support for the cyanobacterial biofilms, which should promote the cyanobacterial inoculum survival and higher efficiency of degraded land restoration based on biocrust carpet engineering.

## Figures and Tables

**Figure 1 polymers-15-02592-f001:**
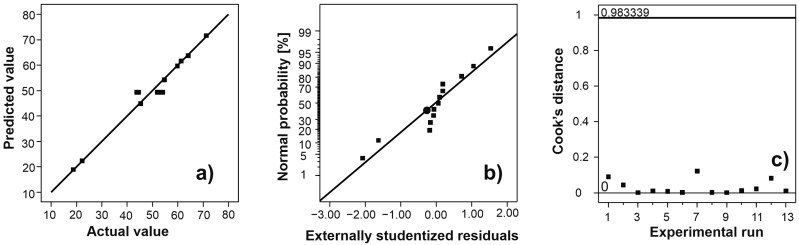
The functionality between predicted and actual values for WRC of alginate hydrogel (**a**), normal probability plot (**b**), and Cook’s distance for the reduced quadratic model (**c**).

**Figure 2 polymers-15-02592-f002:**
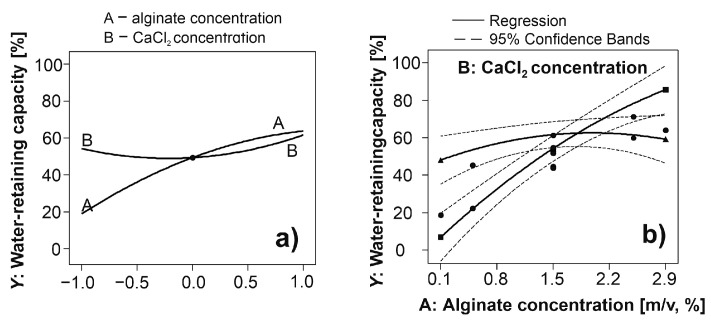
Perturbation plot for comparing the effects of alginate and CaCl_2_ concentrations on the water-retaining capacity (**a**). A is alginate concentration, B is CaCl_2_ concentration, and Y is water-retaining capacity. The functionalities are depicted between water-retaining capacity and coded levels of analyzed factors. The strong interaction effect between alginate and CaCl_2_ concentrations (**b**). The regression and 95% confidence bands are presented as the solid and dash lines, respectively.

**Figure 3 polymers-15-02592-f003:**
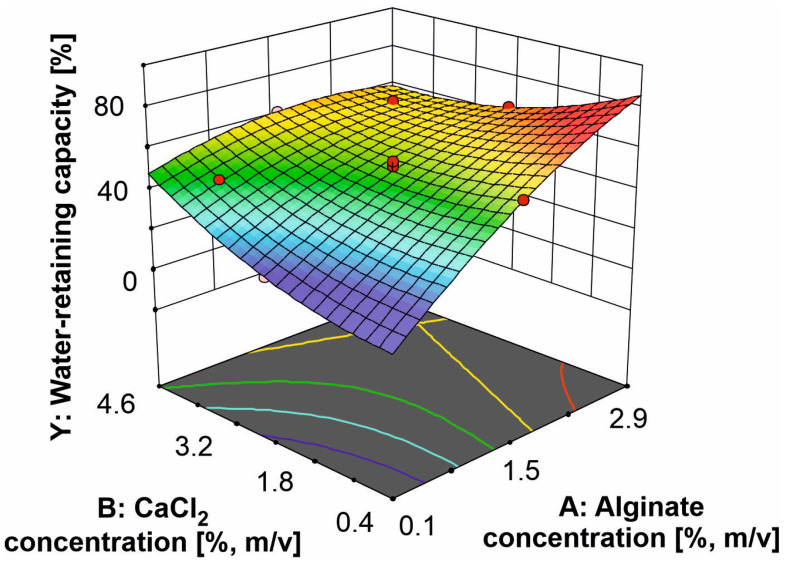
The effect of alginate and CaCl_2_ concentrations on the WRC of alginate hydrogel.

**Figure 4 polymers-15-02592-f004:**
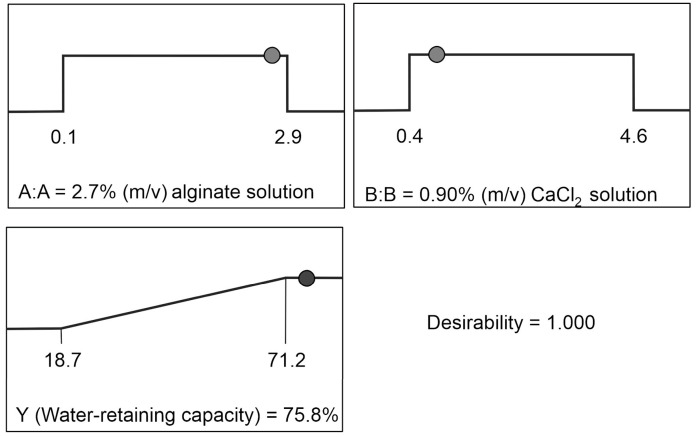
Optimal composition of alginate hydrogel.

**Figure 5 polymers-15-02592-f005:**
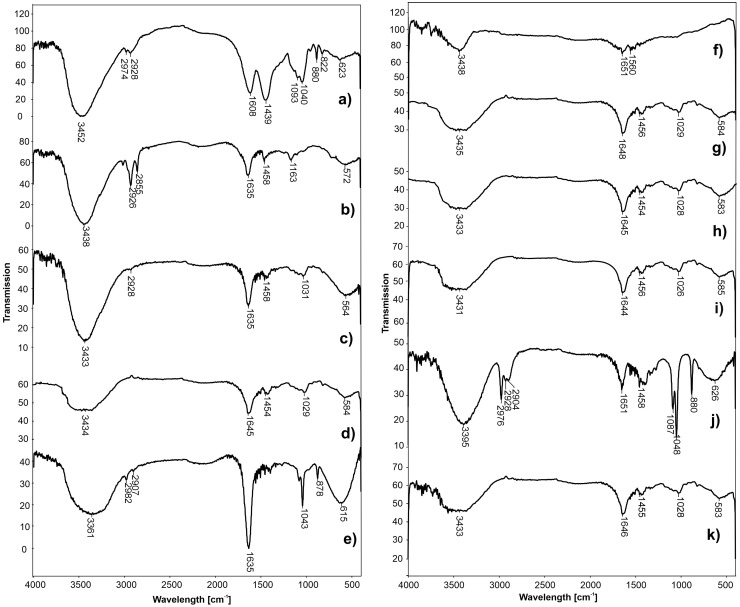
FTIR spectra of Na-alginate (**a**), alginate hydrogel formulations FH1 (**b**), FH3 (**c**), FH4 (**d**), FH5 (**e**), FH6 (**f**), FH8 (**g**), FH9 (**h**), FH10 (**i**), FH13 (**j**), and optimal formulation (**k**).

**Figure 6 polymers-15-02592-f006:**
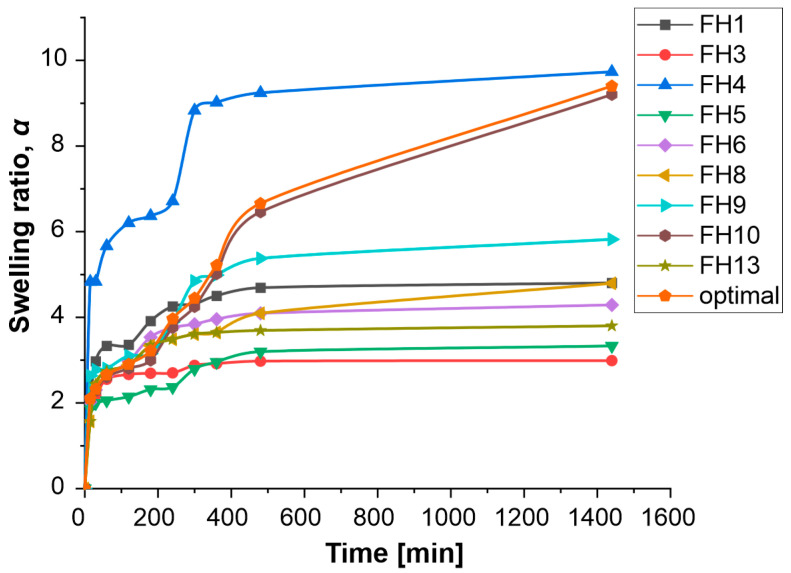
Swelling of alginate hydrogel formulations in distilled water at 25 °C.

**Table 1 polymers-15-02592-t001:** Coded and actual values of factor levels.

Factor/Level (*α*)	−1.414	−1	0	+1	+1.414
A: Alginate concentration (%, m/v)	0.1	0.5	1.5	2.5	2.9
B: CaCl_2_ concentration (%, m/v)	0.4	1.0	2.5	4.0	4.6

**Table 2 polymers-15-02592-t002:** Matrix of CCD with experimental and predicted WRC of alginate hydrogel.

Std.	No.	A: Alginate Concentration (%, m/v)	B: CaCl_2_ Concentration (%, m/v)	WRC of Alginate Hydrogel (%)	
				Exp.	Pred.
9	1 *	1.5	2.5	44.5	49.4
12	2 *	1.5	2.5	52.8	49.4
4	3	2.5	4.0	59.8	59.7
7	4	1.5	0.4	54.7	54.3
8	5	1.5	4.6	61.2	61.6
6	6	2.9	2.5	64.0	63.8
13	7 *	1.5	2.5	43.7	49.4
5	8	0.1	2.5	18.7	19.0
1	9	0.5	1.0	22.4	22.5
2	10	2.5	1.0	71.2	71.6
11	11 *	1.5	2.5	51.8	49.4
10	12 *	1.5	2.5	54.0	49.4
3	13	0.5	4.0	45.3	44.8

WRC—water-retaining capacity, *—design center point.

**Table 3 polymers-15-02592-t003:** Analyzed mathematical models for prediction of WRC of alginate hydrogel.

Equation	Sequential *p*-Value	Lack of Fit *p*-Value	Adjusted R^2^	Predicted R^2^	
Linear	0.0008	0.0992	0.7088	0.4998	
2FI	0.025	0.2163	0.8203	0.7392	
Square	0.0087	0.9977	0.9404	0.9439	Suggested
Cubic	0.9766	0.9933	0.9173	0.9462	Aliased

**Table 4 polymers-15-02592-t004:** Statistical data of fitting for the quadratic model.

Std. Dev.	3.7	R^2^	0.965
Mean value	49.5	Adjusted R^2^	0.940
C.V. %	7.5	Predicted R^2^	0.944
		Adequate precision	20.99

**Table 5 polymers-15-02592-t005:** ANOVA for the reduced quadratic model.

	Sum of Squares	df	Mean Squares	*F*-Value	*p*-Value
Model	2661.91	5	532.38	38.87	<0.0001 *
A—Alginate concentration	2033.93	1	2033.93	148.50	<0.0001 *
B—CaCl_2_ concentration	54.63	1	54.63	3.99	0.0860
AB	297.56	1	297.56	21.72	0.0023 *
A^2^	113.32	1	113.32	8.27	0.0238 *
B^2^	129.35	1	129.35	9.44	0.0180 *
Residual	95.88	7	13.70		
Lack of fit	0.9056	3	0.3019	0.0127	0.9977
Pure error	94.97	4	23.74		
Corrected total	2757.79	12			

*—statistically significant value (*p* < 0.05).

**Table 6 polymers-15-02592-t006:** Gelation time of alginate hydrogel.

Hydrogel Formulation (FH)	Gelation Time (min)	Homogeneity (m_s_/m_v_)				Water Content (%)
		**a**	**b**	**c**	**d**	
FH1	18 ± 1 ^abcd^	0.62	0.62	0.63	0.62	95.06 ± 1.48 ^ab^
FH3	12 ± 1 ^a^	0.54	0.54	0.54	0.55	92.69 ± 0.76 ^a^
FH4	22 ± 4 ^cd^	0.68	0.67	0.68	0.68	95.46 ± 1.09 ^abc^
FH5	14 ± 1 ^ab^	0.52	0.51	0.51	0.53	99.77 ± 0.96 ^d^
FH6	15 ± 2 ^abc^	0.67	0.66	0.67	0.68	99.62 ± 0.61 ^d^
FH8	15 ± 1 ^abc^	0.61	0.60	0.60	0.60	96.38 ± 1.02 ^bc^
FH9	21 ± 3 ^cd^	0.62	0.62	0.62	0.63	98.05 ± 0.78 ^cd^
FH10	17 ± 2 ^abcd^	0.69	0.68	0.69	0.69	93.59 ± 1.32 ^ab^
FH13	19 ± 3 ^bcd^	0.54	0.53	0.54	0.53	95.01 ± 0.87 ^ab^
Optimal	19 ± 1 ^bcd^	0.68	0.69	0.69	0.68	94.26 ± 0.97 ^ab^

Results are presented as mean ± standard deviation (*n* = 3). The identical letters in columns indicated a lack of significant differences at a 95% significance level (Tukey’s HSD test).

## Data Availability

Not applicable.

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
