# Peer review of "Preparation and Characterization of Alginate Hydrogels with High Water-Retaining Capacity"

_polymers, 2023, doi:10.3390/polym15122592_

Round 1
Reviewer 1 Report
1. The alginate-Ca gelation system has been developed amd widely used for more than 30 years. Comparing with traditional characterization method/existing composition, CCD is too complicated. The necessity and advantage of using this method should be discussed.
2. The purpose of the draft is to design a hydrogel system for cyanobacterial growth, minimal biological study should be included.
3. Table 2, the difference between predicted value and experimental value in FH 9 is huge, which needs to be explained. The method to calculate the prediction value should be included in the draft.
4. Line 127, Equation 1 does not make sense for measuring water content.
5. Line 162, Equation 3 is apparently copied and pasted from Equation 2 without appropriate change, which does not match the description in line 162-163.
6. Figure 6, the curves of different group are overlapped with each other at early time points and hard to be distinguished from each other. FH 1,2, 7, 11, 12 has the same formula but the SD differs a lot on 11, which needs to be explained.
7. Line 134-139, the method of measuring gelation time is really unprofessional and not replicable. Rheology method should be considered.
8. Line 173, dynamic viscosity is reported but the characterization method is lacked in the draft.
9. Line 179. confusing expression on the stoichiometry.
10. A lot effort has been put on characterizing/optimizing WRC of the hydrogel system. But the draft is lack of the discussion on its necessity and the scientific design of the calculation equation of WRC as it is not a commonly used factor.
Author Response
REVIEWER 1#
- The alginate-Ca gelation system has been developed and widely used for more than 30 years. Comparing with traditional characterization method/existing composition, CCD is too complicated. The necessity and advantage of using this method should be discussed.
RESPONSE: Thank you for this remark. The CCD approach is maybe complicated but it is very useful. Unlike one factor at a time, the CCD is useful to obtain the global optimal conditions. In this study, it is necessary to obtain the optimal conditions for the preparation of alginate hydrogel which will have a sufficient amount of water required for cyanobacterial growth.
- The purpose of the draft is to design a hydrogel system for cyanobacterial growth, minimal biological study should be included.
RESPONSE: Unfortunately, the cyanobacterial growth is the result that will be included together with other results in the next prepared manuscript. Because of that, it is not suitable representing the results in this manuscript.
- Table 2, the difference between predicted value and experimental value in FH 9 is huge, which needs to be explained. The method to calculate the prediction value should be included in the draft.
RESPONSE: Thank you for this remark. We made a technical error so we corrected it. The adequate value is inserted.
- Line 127, Equation 1 does not make sense for measuring water content.
RESPONSE: We carefully checked the literature (https://doi.org/10.1080/10601325.2017.1294452) and confirmed that our formula for calculating water holding capacity is correct. The marks in our formula now agree with the available literature. Also, this reference includes the part related to the calculation of water retention capacity.
- Line 162, Equation 3 is apparently copied and pasted from Equation 2 without appropriate change, which does not match the description in line 162-163.
RESPONSE: Yes, you are right. We copied and forgot to make the changes in Equation 3. Equation 3 is now corrected.
- Figure 6, the curves of different group are overlapped with each other at early time points and hard to be distinguished from each other. FH 1,2, 7, 11, 12 has the same formula but the SD differs a lot on 11, which needs to be explained.
RESPONSE: We decided to consider the properties of only one hydrogel formulation 1 since the composition of 1, 2, 7, 11, and 12 are the same. It is no sense to lump the data resulting in the unreviewed.
- Line 134-139, the method of measuring gelation time is really unprofessional and not replicable. Rheology method should be considered.
RESPONSE: Thank you for this remark, but this procedure has already been described in the literature for the determination of gelation time. We will have this in mind for our further studies during the determination of hydrogel properties.
- Line 173, dynamic viscosity is reported but the characterization method is lacked in the draft.
RESPONSE: Thank you for this remark, because we did not well describe this part in the manuscript. These properties of alginic acid sodium salt were provided by the manufacturer. Because of that and according to your recommendations, we decided to shift these data to the part of used chemicals and reagents.
- Line 179. confusing expression on the stoichiometry.
RESPONSE: We better explained this part so it is clear now.
- A lot effort has been put on characterizing/optimizing WRC of the hydrogel system. But the draft is lack of the discussion on its necessity and the scientific design of the calculation equation of WRC as it is not a commonly used factor.
RESPONSE: This parameter is probably not a commonly used factor, but it is so extremely important in this study due to providing the hydrogel system with adequate properties necessary for normal cyanobacterial growth.
Reviewer 2 Report
This work prepared a composite hydrogel by copolymerization of alginate and CaCl2 concentrations (factors) on the WRC. However, I don't think this work is suitable for publication on MDPI-Polymers.
1. The preparation of hydrogels using alginate from different sources or its derivatives and copolymer has been widely reported. If only the source of the alginate is changed, I think the innovation of this work is very poor. Because the structure of alginate is basically the same, although the sources are different.
2. The writing format of the manuscript is not standard, and the horizontal and vertical coordinates of the pictures in the article are blurred and illegible, which is even the result of the editor giving the authors a precious opportunity to revise the manuscript carefully. These reflect that the author's attitude towards contributing to MDPI-Polymers is not serious enough.
3. The preparation process of hydrogel is difficult to understand. In the process of reaction at room temperature after adding cross-linking agent and initiator, stirring during the formation of the hydrogel network skeleton will not provide the formability of hydrogel’s structure.
4. Most importantly, as mentioned in the Preparation of alginate hydrogel by ionic gelation and Result & Discussion section by the authors, this work is mainly about the optimization of gelation time, water content, and swelling capacity. The water-retaining & swelling behavior of the hydrogels in this work can reach about 48 & 98, which is impossible. As a high-water content polymer material, the swelling degree of hydrogel is not satisfactory. Some top scientists are also exploring how to improve the swelling degree of hydrogels. So I have great doubts about the authenticity of your results. In addition, the units of swelling degree & water retaining test results, which it difficult to convince people of the authenticity of your work.
5. As for the swelling capacity, the result of the swelling ratio in the work is also puzzling, and there is even no unit in Figure 6. In the “Swelling Behavior", the author wrote that the highest swelling ratio is 98%, which is not satisfactory and is much lower than the current swelling ratio of high-swelling hydrogels. However, according to the "Swelling Ratio", the unit of swelling ratio should be g/g, so I don't know which kind of swelling ratio in this work belongs to. As the main research results of the authors, the results of swelling and water-retaining properties can cause great confusion to others, indicating that the work should not be published. 6. The definition of the physical picture of swelling is poor.
Minor editing of English language required
Author Response
- The preparation of hydrogels using alginate from different sources or its derivatives and copolymer has been widely reported. If only the source of the alginate is changed, I think the innovation of this work is very poor. Because the structure of alginate is basically the same, although the sources are different.
RESPONSE: The topic of our study was not to synthesize new alginate but already the preparation of alginate hydrogel formulation with adequate properties for cyanobacterial growth. Thank you for this idea. It surely will be the topic of our further research.
- The writing format of the manuscript is not standard, and the horizontal and vertical coordinates of the pictures in the article are blurred and illegible, which is even the result of the editor giving the authors a precious opportunity to revise the manuscript carefully. These reflect that the author's attitude towards contributing to MDPI-Polymers is not serious enough.
RESPONSE: We are not sure in which figure you noticed the blurred and illegible coordinates, because they are clear in our document version. Also, we used the journal template for the preparation of our manuscript.
- The preparation process of hydrogel is difficult to understand. In the process of reaction at room temperature after adding cross-linking agent and initiator, stirring during the formation of the hydrogel network skeleton will not provide the formability of hydrogel’s structure.
RESPONSE: I understand your doubt, but it is possible to achieve the preparation of the hydrogel structure during the homogenization process, i.e. stirring on the magnetic stirrer. The part which refers to the hydrogel preparation was improved so that it will be understandable to the readers now.
- Most importantly, as mentioned in the Preparation of alginate hydrogel by ionic gelation and Result & Discussion section by the authors, this work is mainly about the optimization of gelation time, water content, and swelling capacity. The water-retaining & swelling behavior of the hydrogels in this work can reach about 48 & 98, which is impossible. As a high-water content polymer material, the swelling degree of hydrogel is not satisfactory. Some top scientists are also exploring how to improve the swelling degree of hydrogels. So I have great doubts about the authenticity of your results. In addition, the units of swelling degree & water retaining test results, which it difficult to convince people of the authenticity of your work.
RESPONSE: Thank you very much for this recommendation, but the water-retaining and water content were not determined according to the same procedure, i.e. by the same equation. Equation 1 was wrong so we corrected it in the manuscript.
- As for the swelling capacity, the result of the swelling ratio in the work is also puzzling, and there is even no unit in Figure 6. In the “Swelling Behavior", the author wrote that the highest swelling ratio is 98%, which is not satisfactory and is much lower than the current swelling ratio of high-swelling hydrogels. However, according to the "Swelling Ratio", the unit of swelling ratio should be g/g, so I don't know which kind of swelling ratio in this work belongs to. As the main research results of the authors, the results of swelling and water-retaining properties can cause great confusion to others, indicating that the work should not be published. 6. The definition of the physical picture of swelling is poor.
RESPONSE: The water content and swelling degree are not the same parameters. They are determined according to a different procedure. The maximum swelling degree reached a value of over 900%. In Figure 6 and all other parts in the manuscript, the swelling degree was retyped by a swelling ratio (g/g) according to your recommendation.
Reviewer 3 Report
Dear Authors
The optimization of the ionic gelation of alginate hydrogel formation is a classic issue and has been heavily investigated by many researchers from different points of view. The novelty of the presented work is almost absent. The proposed applicability has not been proved.
The following suggestions could improve the quality of the presented work
For instance,
1- The authors did not improve the structure of the alginate with additional hydrophilic groups to increase the water capacity content. Grafting the alginate with additional carboxylate groups could be a solution. For example, modified with iminodiacetic acid to increase the number of carboxylic groups on the glucose ring from one to five.
2- The authors could try the regulation of the ionic gelation process using calcium ions through many approaches as follow;
A) using Iminodiacetic acid calcium ions complex with two free Ca+1 ends instead of calcium ions. This will give the chance to have three dimensions with large pores and so higher water content,
B) Using multi-steps of calcium gelation starting with a very low concentration of calcium chloride, then replaced with a higher concentration.
C) Added a competitive molecule with the calcium ions such as methylene blue in the calcium chloride crosslinked solution, then washed out after different crosslinked times.
3- The authors could try the freeze-drying process to increase the porosity and subsequently the water-holding capacity of the formed alginate hydrogel.
4- Regarding the proposed application of the high water content alginate hydrogel, the authors must provide additional experiments to prove the success of their proposal; "the growth of inoculated cyanobacterial crusts for suppressing the desertification process".
5- The authors need to provide information about the calcium content of each formulation and correlate it to the water-holding capacity.
6- The morphology of the developed alginate hydrogel must be correlated to its water-holding capacity from one side, and to the calcium content from the other side.
Accordingly, I can not recommend the manuscript in its current form for publication.
A minor revision is required.
Author Response
Dear Authors
1- The authors did not improve the structure of the alginate with additional hydrophilic groups to increase the water capacity content. Grafting the alginate with additional carboxylate groups could be a solution. For example, modified with iminodiacetic acid to increase the number of carboxylic groups on the glucose ring from one to five.
2- The authors could try the regulation of the ionic gelation process using calcium ions through many approaches as follow;
- A) using Iminodiacetic acid calcium ions complex with two free Ca+1 ends instead of calcium ions. This will give the chance to have three dimensions with large pores and so higher water content,
- B) Using multi-steps of calcium gelation starting with a very low concentration of calcium chloride, then replaced with a higher concentration.
- C) Added a competitive molecule with the calcium ions such as methylene blue in the calcium chloride crosslinked solution, then washed out after different crosslinked times.
3- The authors could try the freeze-drying process to increase the porosity and subsequently the water-holding capacity of the formed alginate hydrogel.
4- Regarding the proposed application of the high water content alginate hydrogel, the authors must provide additional experiments to prove the success of their proposal; "the growth of inoculated cyanobacterial crusts for suppressing the desertification process".
5- The authors need to provide information about the calcium content of each formulation and correlate it to the water-holding capacity.
6- The morphology of the developed alginate hydrogel must be correlated to its water-holding capacity from one side, and to the calcium content from the other side.
RESPONSE: Thank you for these proposed ideas and recommendations. It will be a great opportunity to analyze the properties of such modified molecules in the next study. Also, the studies that refer to cyanobacterial growth and other results were prepared for another journal because the project plan was to use pure alginate without chemical modification to achieve sustainable development. The project aim is to use juta impregnated with a natural hydrogel with high water content necessary for cyanobacterial growth. Thus prepared material will be used in the real contaminated area.
Round 2
Reviewer 3 Report
Dear Authors
The response to the raised comments is satisfactory.
I have no objection to the publication of the manuscript.
A minor revision of the language may be used.
Author Response
The authors are thankful for the reviewers' recommendation. We carefully checked the language and corrected all noticed grammar errors. All inserted corrections were recorded in the document.
